# Linking the evolution of two prefrontal brain regions to social and foraging challenges in primates

**Sebastien Bouret[1]\*, Emmanuel Paradis[2], Sandrine Prat[3], Laurie Castro[3,4], Pauline Perez[1], Emmanuel Gilissen[5,6], Cecile Garcia[4]**

[1]Team Motivation Brain & Behavior, ICM – Brain and Spine Institute, Paris, France; [2]ISEM, Univ. Montpellier, IRD, EPHE, Montpellier, France; [3]UMR 7194 (HNHP), MNHN/CNRS/UPVD, Musée de l'Homme, Paris, France; [4]UMR 7206 Eco-anthropologie, CNRS – MNHN – Univ. Paris Cité, Musée de l'Homme, Paris, France; [5]Department of African Zoology, Royal Museum for Central Africa, Tervuren, Belgium; [6]Université Libre de Bruxelles, Laboratory of Histology and Neuropathology, Brussels, Belgium

**\*For correspondence:**
sebastien.bouret@icm-institute.
org

**Competing interest:** The authors declare that no competing interests exist.

## eLife assessment

This **important** study correlates the size of various prefrontal brain regions in primate species with socioecological variables like foraging distance and population density. The evidence presented is **solid** but the approach and conclusions are limited to primates with well-defined gyri.

**Abstract** The diversity of cognitive skills across primates remains both a fascinating and a controversial issue. Recent comparative studies provided conflicting results regarding the contribution of social vs ecological constraints to the evolution of cognition. Here, we used an interdisciplinary approach combining comparative cognitive neurosciences and behavioral ecology. Using brain imaging data from 16 primate species, we measured the size of two prefrontal brain regions, the frontal pole (FP) and the dorso-lateral prefrontal cortex (DLPFC), respectively, involved in meta-cognition and working memory, and examined their relation to a combination of socio-ecological variables. The size of these prefrontal regions, as well as the whole brain, was best explained by three variables: body mass, daily traveled distance (an index of ecological constraints), and population density (an index of social constraints). The strong influence of ecological constraints on FP and DLPFC volumes suggests that both metacognition and working memory are critical for foraging in primates. Interestingly, FP volume was much more sensitive to social constraints than DLPFC volume, in line with laboratory studies showing an implication of FP in complex social interactions. Thus, our data highlights the relative weight of social vs ecological constraints on the evolution of specific prefrontal brain regions and their associated cognitive operations in primates.

## Introduction

Numerous studies have addressed the mechanisms underlying the evolution of cognitive abilities in primates, using brain size as a proxy (*Chambers et al., 2021*; *DeCasien et al., 2022*; *Powell et al., 2017*; *van Schaik et al., 2021*). One of the major hypotheses, referred to as the 'social brain' hypothesis, proposes that social interactions require higher cognitive skills such that the complexity of social interactions played a central role in the increase in brain size during primate evolution (*Dunbar, 1998*).

**eLife digest** Primates – such as lemurs, monkeys and humans – can perform a diverse range of cognitive skills, from memory to processing speech. But how did this diversity of cognitive skills evolve?

To answer this question, scientists often compare the brain sizes of different species and measure how this relates to their social behaviors or ecology in the wild. But using the whole brain as a measure of global cognitive capacities seems crude in the light of modern cognitive neuroscience studies, which have shown that specific parts of the brain are responsible for certain cognitive skills.

It is unclear how relevant the findings of cognitive neuroscience studies, which test animals in a laboratory, are to real life situations and evolution. To address this gap, Bouret et al. integrated methods from both behavioral ecology and cognitive neuroscience to examine the size of primate brain regions.

The team studied brain images from 16 primate species, focusing on two regions that have been linked to specific cognitive functions in laboratory experiments. They examined the frontal pole, which is involved in metacognition (the ability to be aware of and assess one's own thoughts), and the dorsolateral prefrontal cortex, which is involved in working memory (the ability to temporarily store information to solve a problem).

Bouret et al. then compared the size of these regions to socio-ecological factors: how much each species performs complex social interactions and how hard it is to find food in the wild, by measuring their population density and daily travel distance, respectively. This revealed that the volume of the frontal pole is larger in species that experience more complex social interactions and in species that struggle to find food – two tasks thought to require metacognition. The dorsolateral prefrontal cortex, however, is only larger in species that have difficulty foraging, which might require a strong working memory to plan travel routes.

These findings suggest that laboratory experiments linking cognitive skills to specific parts of the brain are reliable enough to predict the size of these regions across wild primates. The work of Bouret et al. also helps bridge the gap between cognitive neuroscience and behavioral ecology, and demonstrates how these two disciplines can be combined to investigate the evolution of cognition.

Another major theory, referred to as the 'ecological brain' hypothesis, proposes that the increase in brain size and corresponding cognitive skills was driven by the need to build mental representations in order to forage efficiently (*Milton, 1981*; *Milton, 1993*). Critically, both of these theories are based on the assumption that the whole brain volume is a good proxy for overall cognitive abilities, and that these global abilities can benefit social interactions or foraging, across species in an evolutionary framework. But clearly, the level of precision of such neuro-cognitive scenario remains limited, given the precision with which neuroscience and psychology have characterized the brain functional anatomy and cognition, respectively.

Imaging studies in humans have identified a network of brain regions involved in social interactions, and this network is often referred to as the social brain (*Stanley and Adolphs, 2013*). Recent comparative studies have shown that it also existed in macaques (*Rushworth et al., 2013*; *Sallet et al., 2011*; *Testard et al., 2022*). Along the same lines, laboratory studies have proposed that specific brain regions (e.g. cingulate cortex) were underlying foraging in humans and macaques (e.g. *Hayden et al., 2011*; *Kolling et al., 2012*). These studies, however, have only been conducted in a handful of species and in very artificial conditions, such that the relation between these brain regions and natural behavior or evolution remains elusive. In terms of the cognitive processes involved, 'social interactions' and 'foraging' remain poorly specific and they probably involve myriads of more elementary cognitive operations. For example, social interactions involve some form of categorization of facial expressions and metacognition, and the brain regions associated with these elementary cognitive operations (e.g. frontal pole or temporo-parietal junction) are usually part of the 'social brain', i.e., the set of brain regions associated with social interactions (*de Gelder, 2023*; *de Gelder and Poyo Solanas, 2021*; *Deen et al., 2023*; *Devaine et al., 2017*; *Gallagher and Frith, 2003*; *Rushworth et al., 2013*). Along the same lines, foraging is thought to require spatio-temporal mental representations of food availability and values, and these elementary cognitive operations rely upon a distinct

set of brain regions, including the hippocampus and the ventromedial prefrontal cortex (VMPFC) (*Lin et al., 2015*; *Louail et al., 2019*; *Rosati, 2017*; *Vikbladh et al., 2019*; *Zuberbühler and Janmaat, 2010*). Finally, the brain systems potentially involved in social cognition and foraging partially overlap, especially in the frontal lobes (*Barbey et al., 2014*; *Gallagher and Frith, 2003*; *Kolling et al., 2012*; *Mansouri et al., 2017*; *Rudebeck et al., 2006*; *Yoshida et al., 2012*). This partial overlap between the networks involved in social and foraging cognition in laboratory conditions is in line with the idea that some elementary cognitive operations are generic enough to be involved in multiple contexts (*Garcia et al., 2021*; *Shultz and Dunbar, 2022*). This is typically the case for executive functions (e.g. working memory), which allow to organize behavior over space and time to reach a goal, and rely upon several regions of the prefrontal cortex (*Fuster, 2008*; *Luria, 1973*).

Given the complexity encountered in natural environments, it is hazardous to extrapolate these findings in artificial laboratory conditions to the decision-making processes (in both social and ecological dimensions) occurring in wild primates, for which the diversity of social interactions (e.g. number of individuals an animal can remember, degree of social awareness, i.e. knowledge and representation of the dominance hierarchy, kinship relations, or friendship associations) and foraging behaviors (e.g. food processing techniques, knowledge of harvesting schedules, extractive foraging) require much more cognitive flexibility. One can wonder to what extent these neuro-cognitive concepts (i.e. elementary cognitive operation and their underlying cerebral substrate) can be used as theoretical building blocks to understand the diversity of natural behaviors reported in the wild and their evolution across species. In terms of evolution, the functional heterogeneity of distinct brain regions is captured by the notion of 'mosaic brain', where distinct brain regions could show a specific relation with various socio-ecological challenges, and therefore have relatively separate evolutionary trajectories (*Barton and Harvey, 2000*; *DeCasien and Higham, 2019*). But these relations between specific brain regions and socio-ecological variables across species provide little insight into the cognitive processes that could be at play, because they do not necessarily map onto specific neuro-cognitive processes identified in rigorous laboratory conditions. Thus, it remains challenging to bridge the gap between (1) laboratory studies relating brain regions to elementary cognitive operations in individual species and (2) studies relating brain regions to socio-ecological challenges through an evolutionary scenario (i.e. across numerous species).

In order to bridge this gap, we designed this study to derive and test predictions where a set of well-identified elementary cognitive operations and their associated brain regions (as studied in laboratory conditions) could be involved in more natural functions (as assessed in wild animals living in their natural environments). We focused on two well-known cognitive operations, both considered as executive functions, and involving distinct regions of the prefrontal cortex: metacognition, which involves the frontal pole (FP), and working memory, which involves the dorso-lateral prefrontal cortex (DLPFC), as demonstrated by extensive work in humans and macaques (*Fuster, 2008*; *Mansouri et al., 2017*; *Passingham et al., 2012*). Based on the literature, we hypothesized that FP and metacognition would be associated both with social interactions (by supporting theory of mind, i.e. the ability of an individual to conceptualize others' states of mind) and with foraging (by enabling complex planning) (*Devaine et al., 2017*; *Fleming and Dolan, 2012*; *Frith, 2007*; *Mansouri et al., 2017*). For example, group hunting, which involves both social and foraging functions, only occurs in a few primate species where metacognition is thought to be particularly developed (*Boesch, 1994*; *Conard et al., 2020*; *Garcia et al., 2021*; *Schaik, 2016*). Concerning the DLPFC, given its very clear implication in working memory and planning, we speculated that it would be critically involved in foraging. Its potential relation with social interactions, however, was less clear: on one hand, DLPFC is rarely associated with social interactions in laboratory studies (*Frith, 2007*; *Fuster, 2008*; *Passingham et al., 2012*; *Sallet et al., 2013*). But, on the other hand, working memory and planning could readily be involved in complex social interactions in more natural conditions.

To evaluate the relevance of laboratory-based neuro-cognitive concepts for understanding which cognitive operations could be mobilized to overcome specific socio-ecological challenges in the wild and their variability across species, we addressed the following central question: *to what extent does the functional mapping between brain regions and specific executive functions in laboratory conditions relate to variability in socio-ecological variables at the interspecific level?* For this purpose, it is necessary to go beyond the handful of species usually studied in well-controlled laboratory conditions,

not only to evaluate the reliability of the relations between neuro-anatomical measures and socio-ecological factors across primates, but also to quantify the contribution of phylogeny.

To address this question, we measured the volume of the two prefrontal regions of interest (FP and DLPFC) in 16 primate species and evaluated the influence of key socio-ecological factors on the size of these regions. Based on the known positive relation between the size of a given brain region (or the number of corresponding neurons) and the relative importance of its associated function, both within and across species, we assume that the size of each brain region can be taken as a proxy for the weight of its associated function on behavior (*Barks et al., 2015*; *Barton and Harvey, 2000*; *DeCasien and Higham, 2019*; *Ferrucci et al., 2022*; *Herculano-Houzel, 2017*; *Herculano-Houzel et al., 2016*; *Louail et al., 2019*; *Maguire et al., 2000*; *Sallet et al., 2011*; *Testard et al., 2022*). Anatomical studies have shown that these regions were relatively well conserved between humans and macaques, both in terms of cytoarchitectonics and connectivity profiles (*Amiez et al., 2019*; *Petrides et al., 2012*; *Sallet et al., 2013*). Here, these two regions were chosen and identified based on functional maps and reliable macroscopic landmarks, rather than cytoarchitectonic criteria, to maximize the reliability of the comparative approach (see also *Louail et al., 2019*). Finally, the cognitive functions in which they are involved appear relatively well conserved between macaques and humans (*Ferrucci et al., 2022*; *Fuster, 2008*; *Mansouri et al., 2017*; *Passingham and Sakai, 2004*). The 16 primate species were chosen to cover a wide range of phylogenetic distances coupled to a diversity of socio-ecological variables. We compared the influence of several combinations of these socio-ecological variables on the variability in size of each brain region of interest, and interpreted it in the light of its known function in laboratory conditions. Thus, for a given functional region, the differences in regional volume across species provide a reliable index of the differences in neuronal count, and therefore of the skill level in the corresponding function. Even though comparative studies often used scaled measures (i.e. relative to the whole brain) to account for potential allometric effects, we favored absolute measures because scaling procedures would distort the relation between volume and neural counts (*Barton et al., 1995*; *DeCasien et al., 2022*; *Gabi et al., 2016*; *Herculano-Houzel, 2017*; *Herculano-Houzel et al., 2008*; *Krebs et al., 1989*; *Smaers et al., 2017*; *Smaers and Soligo, 2013*). Moreover, the extent to which other brain regions could be affected by socio-ecological variables, and therefore affect scaled measures, is difficult to evaluate. We therefore used several procedures to evaluate the specificity of the effects of socio-ecological factors on FP and DLPFC. First, we compared the influence of socio-ecological variables not only on FP and DLPFC but also on the whole brain. Second, we included the whole brain volume as a covariate along with the socio-ecological variables to account for FP and DLPFC volumes. Altogether, these procedures provide a reliable indication of the specificity of the influence of socio-ecological variables on the FP and DLPFC volumes, used as proxies for inter-species differences in metacognitive and working memory skills, respectively. used as proxies for inter-species differences in metacognitive and working memory skills. Thus, we could

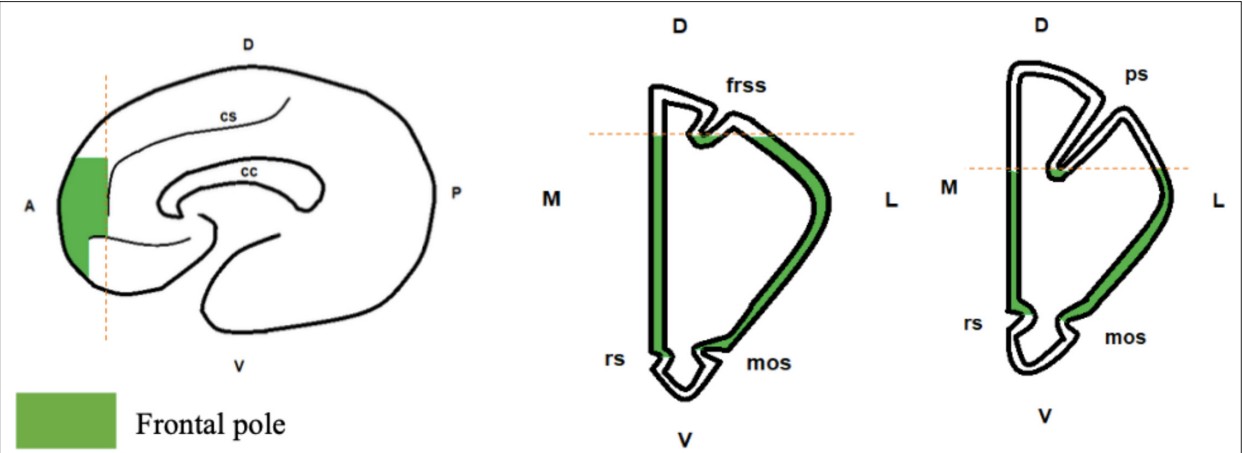

**Figure 1.** Boundaries of the frontal pole. From left to right: sagittal view, coronal view for great apes and humans, coronal view for monkeys. Frontal pole is in green. Abbreviations: cs: cingulate sulcus; cc: corpus callosum; rs: rostral sulcus; mos: medial-orbital sulcus; frss: superior frontal sulcus; ps: principal sulcus; D: dorsal; V: ventral; M: medial; L: lateral; A: anterior; P: posterior.

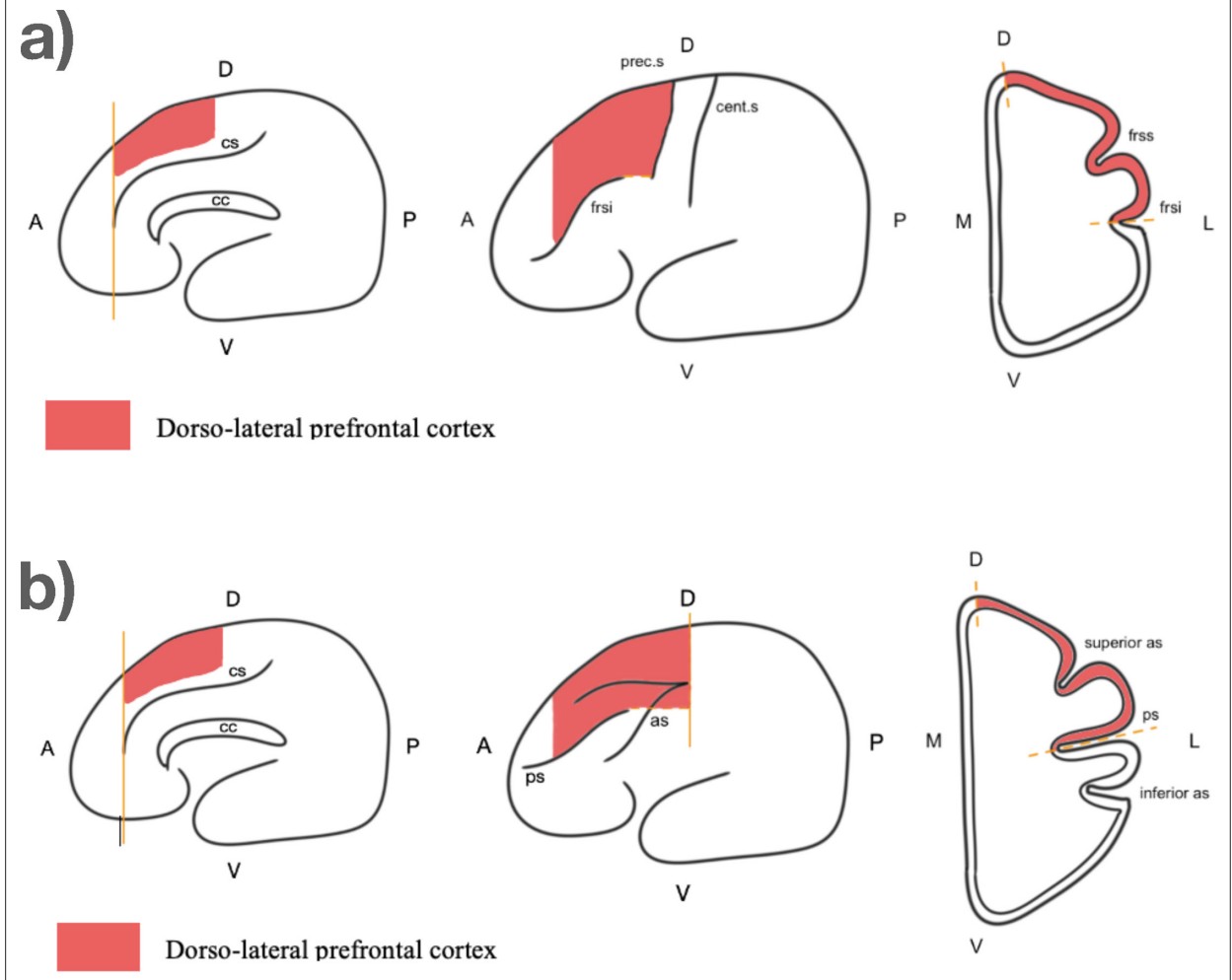

**Figure 2.** Boundaries of the dorso-lateral prefrontal cortex. (**a**) For great apes and humans. From left to right: sagittal view, external view, coronal view. Abbreviations: cs: cingulate sulcus; cc: corpus callosum; frsi: frontal inferior sulcus; frss: frontal superior sulcus; cent.s: central sulcus; prec.s: precentral sulcus; A: anterior: P: posterior; D: dorsal; V: ventral; L: lateral; M: medial. (**b**) For monkeys. From left to right: sagittal view, external view, coronal view. Abbreviations: cs: cingulate sulcus; cc: corpus callosum; ps: principal sulcus; as: arcuate sulcus; A: anterior: P: posterior; D: dorsal; V: ventral; L: lateral; M: medial.

evaluate the strength of the relation between specific regions of the prefrontal cortex and specific socio-ecological variables across species, as observed in the wild, thereby complementing laboratory studies and bridging the gap between cognitive neurosciences, behavioral ecology, and primate evolution.

## Results

### Neuroanatomical measures

We measured the volume of the FP in 31 brains from 16 species. The boundaries of the FP (*Figure 1*) were chosen based on a combination of functional and anatomical criteria from the literature (see Materials and methods for details).

We measured the volume of the DLPFC in 31 brains from 16 species. The boundaries of the DLPFC (*Figure 2*) were chosen based on a combination of functional and anatomical criteria from the literature. By contrast with the FP, we had to use distinct landmarks for monkeys vs humans and great apes (see Materials and methods for details).

The average size of the regions of interest (whole brain, FP, and DLPFC) are shown on *Figure 3*. As expected, all these measures were highly correlated (all r=0.99, Pearson correlation). Not only the

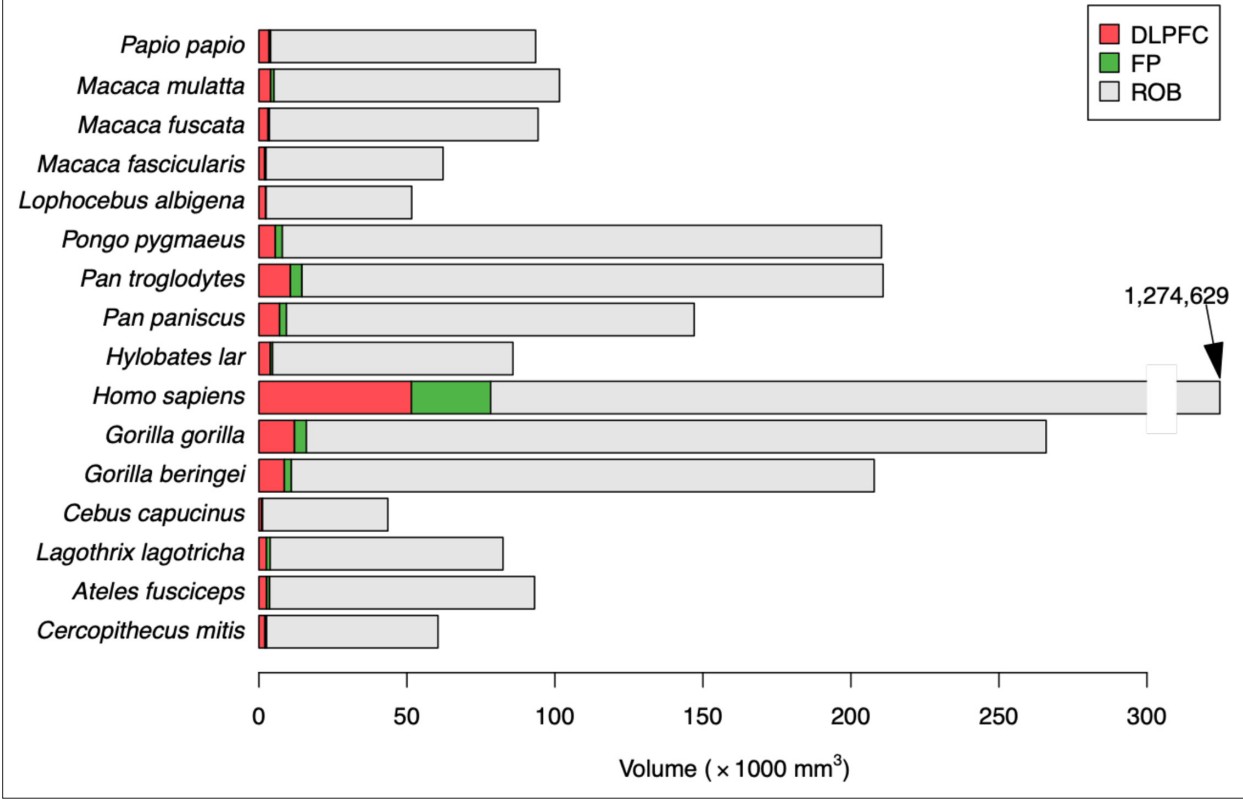

**Figure 3.** Average values of the three regions of interest. Each line provides the cumulated volumes of the dorso-lateral prefrontal cortex (DLPFC, green), the frontal pole (FP, gray), and the rest of the brain (ROB, blue), such that the size of each bar represents the whole brain volume. Note: for *Homo sapiens*, the bar has been truncated, since its value was out of scale with the other species.

volumes of DLPFC and FP, which are two neighboring regions of the prefrontal cortex, but also each of these regions and the whole brain.

### Influence of socio-ecological variables on whole brain volume

We used a model comparison approach to select the best combination of socio-ecological variables accounting for the variability in whole brain size across species using phylogenetic generalized least squares (PGLS). Details of the different models are provided in *Supplementary file 1*.

The best model explaining the volume of the whole brain is the one that includes body mass, daily traveled distance, and population density. The values of the estimated coefficients are provided in *Table 1*. These three variables have a positive influence on the volume of the whole brain. Even if all three factors had a significant effect, their influence on the whole brain volume seemed to differ: there was an order of magnitude between the estimated beta (i.e. contribution) of body mass (0.47±0.15) and that of daily traveled distance (0.05±0.01). There was also an order of magnitude between the estimated beta of daily traveled distance and population density (0.007±0.003). Note that the standardized beta (see Materials and methods) is also greater for daily traveled distance than for population density (see *Supplementary file 2*).

**Table 1.** Estimated coefficients of socio-ecological variables for the whole brain volume.

| Predictor | Beta (estimate) | Beta (std. error) | Standardized beta | t-Value | p- Value |
|---|---|---|---|---|---|
| Intercept | 4.26 | 0.24 | | 17.8 | 0 |
| Body mass (log) | 0.47 | 0.15 | 0.71 | 3.10 | 0.009 |
| Daily traveled distance | 0.05 | 0.01 | 0.42 | 4.08 | 0.002 |
| Population density | 0.007 | 0.003 | 0.35 | 2.35 | 0.036 |

**Table 2.** Estimated coefficients of socio-ecological variables for the frontal pole volume.

| Predictor | Beta (estimate) | Beta (std. error) | Standardized beta | t-Value | p.-Value |
|---|---|---|---|---|---|
| Intercept | 1.87 | 0.37 | | 5.08 | 0 |
| Body mass (log) | 0.69 | 0.23 | 0.66 | 2.96 | 0.012 |
| Daily traveled distance | 0.07 | 0.02 | 0.36 | 3.62 | 0.004 |
| Population density | 0.011 | 0.004 | 0.38 | 2.62 | 0.022 |

We evaluated the robustness of the influence of these socio-ecological variables on whole brain volume using a 'leave-one-out' procedure. Thus, each model was tested 16 times (one per species removed) and we evaluated the reliability of each variable after each species was removed. The details of this analysis are provided in *Supplementary file 3*. The influence of body mass was very robust, in that it remained significant for all models where one species was removed. Daily traveled distance remained significant for all models but one, whereas population density lost significance ($p > 0.05$) for four models where one species was removed, even if in all cases the estimated coefficient remained positive. Thus, the influence of body mass and daily traveled distance are very robust for the combination of species used for the analysis. It is less the case for population density, even if based on model comparison this variable plays a significant role in increasing the volume of the whole brain.

In summary, this analysis indicates that the volume of the whole brain across primates is positively modulated by body mass, as well as social (population density) and ecological (daily traveled distance) variables. The weight of the ecological variable, however, seems to be slightly stronger and more reliable than the weight of the social variable.

## Influence of socio-ecological variables on FP volume

As for the whole brain, the best model accounting for the volume of the FP is the one that includes body mass, daily traveled distance, and population density. Details of the different models are provided in *Supplementary file 1*. The values of estimated coefficients, all positive, are provided in *Table 2*. Note that, as expected given the strong correlation between FP and whole brain volumes, including the whole brain volume as a covariate in the regression models canceled the effects of all other variables, such that only whole brain showed a significant effect.

As for the whole brain, the largest effect is body mass ($0.69 \pm 0.23$), then daily traveled distance ($0.07 \pm 0.02$) and population density ($0.011 \pm 0.004$). By comparison with the whole brain, however, the coefficients for daily traveled distance and population density have the same order of magnitude such that their impact on the volume of the FP seems to be equivalent. Note that the standardized beta for daily traveled distance is smaller than for population density, whereas it was the opposite for the whole brain, i.e., bigger for daily traveled distance compared to population density.

As indicated in the Materials and methods section, we also conducted an analysis using a less conservative evaluation of FP volumes in monkeys. As shown in *Supplementary file 4*, the relative weight of population density decreased when adding cortical regions dorsal to the principal sulcus. Thus, when using a less conservative estimate of the FP volume, its relation with socio-ecological variables was closer to that of the whole brain.

As for the whole brain, we evaluated the reliability of the model's variables using a 'leave-one-out' procedure at the species level. The results of this analysis were similar to what we reported for the whole brain: the influence of body mass remained significant whatever the species removed, only one species caused daily traveled distance to lose significance and three species caused population density to lose significance ($p > 0.05$). As can be seen in *Supplementary file 5*, however, the decrease in the weight of the estimated coefficient for population density was less dramatic for these three species, compared to what we observed for the whole brain. As specified in the Materials and methods section, we also conducted an analysis where the dorsal limit of the FP was less conservative, to include a part of the brain (BA 9) which might belong to the FP but also to the DLPFC. Given the difficulty to evaluate more precisely the border between DLPFC and FP, especially in species for which there is no brain atlas, we favored the conservative definitions of the FP and the DLPFC, and excluded this dorsal area which either includes a mixture of brain regions specifically associated with the DLPFC or the FP, or could be defined as an intermediate functional region between the two. The detailed

**Table 3.** Estimated coefficients of socio-ecological variables for the dorso-lateral prefrontal cortex (DLPFC) volume.

| Predictor | Beta (estimate) | Beta (std. error) | Standardized beta | t-Value | p-Value |
|---|---|---|---|---|---|
| Intercept | 2.78 | 0.28 | | 9.99 | 0 |
| Body mass (log) | 0.47 | 0.18 | 0.62 | 2.67 | 0.021 |
| Daily traveled distance | 0.047 | 0.014 | 0.35 | 3.3 | 0.006 |
| Population density | 0.006 | 0.003 | 0.30 | 2.01 | 0.067 |

results of the analysis using the less conservative definition of the FP are shown in **Supplementary file 4**. In short, when the dorsal area was included in the FP volume, the influence of the social variable (population density) decreased, compared to the more conservative definition of FP. This is in line with the idea that the dorsal area could also be associated with the DLPFC, given the greater influence of population density on the FP (with conservative limits) than on the DLPFC.

Altogether, these data indicate that the volume of the FP across primates is positively modulated by body mass as well as by both population density and daily traveled distance, in line with the idea that it is affected by both social and ecological components. By comparison with the whole brain, the relative influence of the social (population density) and ecological (daily traveled distance) variables are more balanced.

### Influence of socio-ecological variables on DLPFC volume

As for the whole brain and the FP, the best model for the DLPFC is the one that includes body mass, daily traveled distance, and population density. Details of the different models are provided in **Supplementary file 1**. The values of estimated coefficients, all positive, are provided in **Table 3**. As it was the case for the FP, including the whole brain volume as a covariate in the regression models canceled the effects of all other variables, such that only whole brain showed a significant effect.

As for the whole brain and the FP, the largest effect is body mass (0.47±0.18), followed by daily traveled distance (0.047±0.01). Even if the best model for DLPFC includes population density, its influence was very small (0.006±0.003) and failed to reach significance (p=0.067). The 'leave-one-out' procedure confirmed the reliability of the model for body mass (with no species causing that variable to lose significance when removed) and that of daily traveled distance (only one species causing that variable to lose significance when removed). The influence of population density only reached significance for two of the models where one species was removed, but it failed to reach significance for all other combinations tested (n=14), as it was the case for the original one with all 16 species. Details of this analysis can be found in **Supplementary file 6**. In other words, the relative weakness of the influence of population density compared to that of other variables is relatively reliable across the combinations of species used for the tests.

In summary, the volume of the DLPFC across primates is also positively modulated by body mass as well as by both ecological (daily traveled distance) and social (population density) variables, but the influence of the later appears much weaker than that of the former.

## Discussion

We evaluated the influence of 11 socio-ecological variables on the size of the whole brain as well as two specific brain regions involved in executive functions: the FP and the DLPFC. Altogether, our results showed that all three cerebral measures strongly correlated and were influenced by the same set of socio-ecological variables: body mass, daily traveled distance, and population density. As expected from evidence suggesting brain-body covariation (**Martin, 1981**), body mass had a strong and reliable influence on all brain measures. Daily traveled distance, a proxy for how challenging is foraging, also had a clear positive influence on all brain measures. Finally, population density, a proxy for how challenging and complex are social interactions, was generally less powerful at explaining neuro-anatomical variability, but it was also the variable showing the greatest difference in effect size across brain regions. For the FP, the influence of population density was similar to that of daily traveled distance, but it was much weaker for the whole brain, and it even failed to reach significance for the DLPFC. Thus, our data are generally compatible with the idea that the evolution of executive functions

relying upon FP and DLPFC is driven both by ecological and by social constraints. Critically, the relative influence of these constraints seems to vary across regions, in line with our hypothesis based on the cognitive functions with which they are associated in laboratory conditions. Thus, neuro-cognitive entities as defined in laboratory conditions can readily be articulated with socio-ecological processes in the wild, in order to better understand primate cognition and behavior in natural conditions.

As we and others have done before (within and across species), we assume that the size of a given brain region provides a good proxy of the strength of its influence (through its known cognitive function) on behavior (*Barton and Harvey, 2000*; *DeCasien and Higham, 2019*; *Louail et al., 2019*; *Maguire et al., 2000*). Indeed, the size of a brain region provides a reliable estimate of the number of neurons allocated to its function, given the known positive relation between number of cortical neurons and cognitive skills in primates (*Herculano-Houzel, 2017*; *Herculano-Houzel, 2018*). Here, we restricted our analysis to a set of specific brain regions, namely the FP and the DLPFC, assuming that the difference in size of these brain regions across species could readily be used as proxies for the relative importance of their corresponding functions (namely, metacognition for the FP and working memory for the DLPFC). Using that tool, we explored the relation between neuro-cognitive entities (e.g. FP/metacognition vs DLPFC/working memory) and socio-ecological variables, based on the assumption that the more a given cognitive function would be required to face a given socio-ecological challenge, the bigger the corresponding brain region would be.

Our aim here was clearly not to provide an accurate identification of anatomical boundaries across brain regions in individual species, as others have done using finer neuro-anatomical methods in humans and macaques (e.g. *Petrides et al., 2012*). Moreover, we cannot affirm that our conclusions in gyrencephalic species could be extended to lissencephalic species. Indeed, we acknowledge that since the definition of the regions of interest we used is based on gyri and sulci, it was therefore impossible to include lissencephalic primates in our sample, even if previous laboratory studies in marmosets could identify DLPFC and FP using both cytoarchitectonic and functional criteria (*Dias et al., 1996*; *Dureux et al., 2023*; *Roberts et al., 2007*; *Wong et al., 2023*). Nonetheless, since our comparative study and the associated phylogenetic analysis required at least 10–15 species, it could not have been conducted with only laboratory species and these invasive approaches, and we therefore focused on gyrencephalic species.

The mapping between the regions measured here and either the specific cognitive operations of interest (metacognition vs working memory) or the socio-ecological variables (e.g. social or foraging complexity) is probably not exclusive. First, we are aware of the complexity of identifying neural processes underlying such high-level cognitive functions, and even in well-controlled laboratory conditions, the delimitation of exact boundaries of cortical regions remains challenging (*Mansouri et al., 2017*; *Petrides, 2005*; *Sallet et al., 2013*). Rather, we tried to maximize the reliability of the landmarks that could be identified in all species to compare the relative size of areas that are essentially used as proxies for their known function, as defined in laboratory conditions in macaques and humans. The underlying assumption that we used was that the functional anatomy of the prefrontal cortex was enough conserved in primates, such that traits that were common between humans and macaques were also shared with other primates (*Amiez et al., 2019*; *Bludau et al., 2014*; *Sallet et al., 2013*). Of course, the neural mechanisms underlying metacognition and working memory are extremely complicated and they involve myriads of other brain regions (e.g. temporal and parietal cortices, as well as subcortical systems). Moreover, the exact boundary between FP and DLPFC as functional entities remains difficult to assess, even in laboratory conditions (*Boorman et al., 2009*; *Gallagher and Frith, 2003*; *Genovesio et al., 2014*; *Koechlin, 2016*; *Passingham and Sakai, 2004*; *Preuss and Wise, 2022*). But the contribution of the FP to metacognition and of the DLPFC to working memory is so strong and reliable in humans and laboratory primates that it seems reasonable to assume that, given the relative level of conservation of prefrontal cortex anatomical organization across primates, FP and DLPFC should also play a key role in metacognition and working memory in other primate species. As for our previous study using a similar approach, this method was reliable enough to capture meaningful and specific effects of interest (*Louail et al., 2019*). Thus, even if we make no strong claim regarding the specific boundaries of FP and DLPFC, the anatomo-functional difference between these regions, as characterized in a few species used in laboratory experiments, is reliable enough to be captured by the variability in socio-ecological variables in a larger set of primates.

In previous studies, the specificity of the relation between a given brain region and socio-ecological variables has often been evaluated using relative measures, such that the variable represents the relative increase in volume of the region relative to the rest of the brain (e.g. *Krebs et al., 1989*; *Barton et al., 1995*). This approach enables to detect changes in relation to allometric relations, which are often interpreted in terms of significant evolutionary events (major reorganization of the brain). By contrast, changes in regional volume that remain within allometric proportions (in the brain) are interpreted as a non-specific effect, and therefore negligible from an evolutionary perspective (*Barton and Harvey, 2000*; *Barton et al., 1995*; *DeCasien et al., 2022*; *Krebs et al., 1989*; *Smaers et al., 2017*). But even if we share the intuition of specificity captured by relative measures, we decided to use absolute measures because relative measures have several theoretical implications that we find problematic. First, relative measures imply that the effect of interest should be negligible in the rest of the brain, compared to the area of interest. But we made no predictions about the rest of the brain given that most cognitive operations involve networks rather than unique regions. For example, since working memory also relies upon parietal cortices, we expect parietal cortices to also show a positive relation with ecological constraints. Thus, the theoretical effect size of a relative measure, given the functional specificity hypothesis, would require identifying the relative volume of all regions involved in working memory (for DLPFC) and metacognition (for FP), which is clearly beyond the scope of this study. As it will be discussed in the next paragraph, we believe that the comparison between DLPFC, FP, and whole brain measures provides a good index of specificity of the effects, and they can be interpreted more directly. Second, the interpretation of relative measures implies that, all other things being equal, a pool of neurons involved in a given function is less efficient when surrounded by a large number of neurons, compared to the situation where the same pool is surrounded by fewer neurons. For example, the hippocampus occupies a much larger fraction of the brain in rodents compared to primates, such that based on a relative size measurement, one would conclude that rodents have better spatial and episodic memory skills compared to primates. And even if the comparison remains difficult, there is clearly no evidence for that in behavioral data (*Crystal, 2009*; *de Cothi et al., 2022*). In summary, even if we acknowledge that relative measures provide an index of the specificity of the relation between socio-ecological variables and regional brain volumes, we find relative measures much more difficult to interpret and we chose to use absolute measures of cerebral volumes to stay away from these considerations. Indeed, the number of neurons in a given brain region should increase if its function can promote survival, irrespective of what happens in other regions.

We did evaluate the specificity of the effects by comparing the influence of socio-ecological variables not only between the two specific prefrontal regions but also with the whole brain volume. At first sight, the very strong correlation between the volume of the whole brain and that of both FP and DLPFC implies that the specificity of the influence of socio-ecological factors on these prefrontal regions should be limited, and indeed, the variability of all three measures (FP, DLPFC, and whole brain) is predicted by the same model, i.e., the same combination of socio-ecological variables. This is in line with the idea that executive functions are strongly inter-related and it can prove difficult to demonstrate their independence, even within a single species in controlled conditions (*Miyake et al., 2000*; *Völter et al., 2022*). But in spite of this strong correlation and in line with our prediction based on laboratory studies, we could still find significant differences between FP and DLPFC in terms of relation with population density, an index of social complexity.

How reliable is the difference we reported between brain regions? Even if the same combination of variables accounted for the relative size of all three brain measures, there was a difference in the weight of population density, proxy for social challenges, across the three regions. Indeed, for the FP, the effect of population density was as strong as that of daily traveled distance: both were significant, and their effect size (beta weight) was of the same order of magnitude. Also, they showed a similarly small sensitivity to the 'leave-one-out' procedure. Finally, standardized beta weights for population density were greater than for daily traveled distance. By contrast, for the DLPFC, the effect of population density failed to reach significance (p=0.07), its beta weight was one order of magnitude smaller than that of daily traveled distance and standardized beta weights for population density were smaller than for daily traveled distance. Finally, this relatively weak influence of population density on DLPFC volume was confirmed by the 'leave-one-out' procedure, since the effect of population density on the size of the DLPFC was reliably marginal across all combination of species used to fit the model. Interestingly, the whole brain volume (which is strongly correlated with both FP and DLPFC) seems to

show an intermediate tendency: even if the effect of population density appears smaller than that of daily traveled distance (beta weight is one order of magnitude smaller), it is clearly significant, and the 'leave-one-out' procedure indicated that it was only slightly less reliable than daily traveled distance, with a significant decrease in model fit quality for four vs one combination of species. Altogether, this indicates that even if the influence of population density cannot be ruled out for any of the three brain measures, it appears quantitatively smaller for the DLPFC than for the FP. Note that the relative weight of population density also decreased (relative to that of daily traveled distance) when we used a less conservative estimate of FP volumes, which included a dorsal part potentially overlapping with the DLPFC. This is also compatible with the fact that the influence of population density seems to be of intermediate magnitude in the whole brain, which includes both FP and DLPFC. Again, we acknowledge that this method has limitations and that further studies, including more species, would be necessary to evaluate the nature and the dynamics of underlying evolutionary processes.

To what extent do these conclusions depend upon the specific sample of species used here? The results of the 'leave-one-out' analysis showed that our sample, even if relatively limited, was sufficient to evaluate the relative weight of socio-ecological variables on specific brain regions in primates. Interestingly, the relation between brain region volumes and socio-ecological variables was not completely independent from the sample of species, as previously shown in a study conducted on a larger sample (*Powell et al., 2017*). In line with the main analysis (PGLS with all species included), the influence of body mass and daily traveled distance appears very reliable (no more than one species caused the model to fail when removed). The influence of population density appeared less reliable, since it was sensitive to the removal of up to three to four species (as a function of brain region, see previous paragraph). But the influence of individual species is nearly impossible to interpret, because no apparent obvious pattern emerges. Indeed, the species without which the model failed was not always the same (e.g. humans and baboons have very distinct values for population density, 56 vs 7.5). Note, however, that getting clear intuitions from the data remains difficult because the PGLS includes the covariance induced by phylogeny to the linear relation across variables. We see no reason to exclude humans (as done in previous studies, e.g. *DeCasien and Higham, 2019*), because even if the brain of *H. sapiens* is much bigger than others, a general biological law describing the relation between socio-ecology and the brain in primates should also apply to humans (*Gabi et al., 2016*; *Herculano-Houzel, 2017*). Moreover, even if the human brain is much bigger than that of other primates, the global organization of the frontal cortex appears qualitatively similar in humans and other primates (*Barrett et al., 2020*; *Gabi et al., 2016*; *Gabi et al., 2016*; *Roumazeilles et al., 2020*; *Sallet et al., 2013*). One potential issue with humans is the difficulty to evaluate socio-ecological variables, given that modern human populations show a tremendous geographical variability in terms of socio-ecological variables. But the same problematic applies to other modern primate species (including those confronted to intense anthropization of their habitat) (*McKinney et al., 2023*). Thus, the actual potential limitation (which concerns all species) is the reliability with which socio-ecological variables were estimated for each species, given the amount of intra-specific variability. Note, there is also a significant amount of intra-specific variability at the level of the brain, both in humans and in non-human primates, but this intra-specific variability was shown to be negligible compared to inter-specific variability (*DeCasien and Higham, 2019*; *Louail et al., 2019*; *Maguire et al., 2000*; *Testard et al., 2022*). Thus, we acknowledge that intra-specific variability could introduce noise in the inter-specific relation between brain measures and socio-ecological variables, but the fact that clear relations could be established with our sample indicates that this source of intra-specific variability was limited enough relative to the inter-species relation between neuroanatomical measures and socio-ecological variables. In other words, the error with which these variables were estimated at the level of individual species was small enough to allow us to study the relation of interest across species, and from that perspective, there is no reason to exclude any species from that analysis and our sample appears reliable enough to characterize the relation between neuroanatomical features and socio-ecological variables across primates.

The stronger sensitivity of the FP to the variable 'population density' is reminiscent of the social brain hypothesis, i.e., the idea that social challenges favored the evolution of larger brains, and especially larger neocortex size, with species living in larger groups having bigger brains to deal with the associated complex social interactions (*Dunbar, 1998*). Critically, the size of the FP and the corresponding development of metacognitive skills is not exclusively related to population density (proxy for social interactions) but also to daily traveled distance (proxy for foraging complexity). Thus, these data are

also compatible with the ecological brain hypothesis (*Milton, 1981*). This dual relation between FP size, social, and ecological constraints might be accounted for by the fact that in the wild, social and ecological constraints remain strongly related, even if we did not find strong correlations between those factors in our data set. In reality, it is difficult to treat ecological and social factors as if they were disconnected (*Henke-von der Malsburg et al., 2020*), and cognitive abilities associated with foraging are likely to play a role in social foraging tactics too (*Street et al., 2017*). Indeed, greater population density implies a greater inter-individual competition for food, and thus potentially increases in daily traveled distances, and/or the development of sophisticated foraging skills to deal with the increase in scramble competition. But from a cognitive point of view, this also implies that the benefits associated with increased FP volume, and presumably an increase in metacognitive skills, could be related to both social and ecological functions. Indeed, as pointed out earlier, the development of executive functions in general, and metacognitive skills in particular, could have been a critical leverage to allow the development of both complex social interactions (through the use of theory of mind) and complex foraging (through flexible, context-dependent planning) (*Garcia et al., 2021*; *Shultz and Dunbar, 2022*). Thus, our work indicates that the notions of 'social brain' and 'ecological brain' should not be mutually exclusive, even when considering specific brain regions. Rather, we confirm intuitions based on laboratory studies that the FP, through its role in metacognition, belongs to both sets of brain regions defined as the social brain and the ecological brain, respectively. Further work would be needed to capture the metacognitive processes underlying social and foraging processes in the wild, as well as their interactions.

The size of the DLPFC, which we used as a proxy for working memory and planning skills, showed a significant relation with daily traveled distance (proxy for foraging complexity), and to a lesser extent with population density (proxy for social complexity). This suggests that the cognitive functions at play in the DLPFC, i.e., working memory and planning, are critical for foraging in primates, especially when they need to travel long distances, and presumably have to deal with more complex navigation strategies. This is clearly in line with the global idea of the ecological brain hypothesis, but here we provide a critical insight into the specific neuro-cognitive operations associated with foraging strategies. The weaker influence of population density, a marker of social complexity, might be surprising at first sight, because a priori working memory and planning could also be strongly involved in complex social interactions (*Garcia et al., 2021*). But it is in line with laboratory data showing that social interactions seem to rely much more upon rostro-medial prefrontal regions compared to DLPFC (*Fleming and Dolan, 2012*; *Frith, 2007*; *Sallet et al., 2011*; *Testard et al., 2022*). In other words, this work confirms the specificity of the 'social brain' as defined in laboratory conditions as a set of structures specifically involved in social interactions (*Frith, 2007*; *Rushworth et al., 2013*). Critically, again, social and ecological functions are tightly intermingled in primates' natural environment and more specific studies would be needed to clarify how DLPFC-related functions such as working memory are involved in natural conditions, when animals need to face both ecological and social challenges.

This new set of results can be integrated in the general framework of the primate mosaic brain evolution, i.e., the different distinct structures varying in size both within and between species, and reflecting selection for cognitive skills. In line with recently published papers (*DeCasien et al., 2022*; *Smaers et al., 2021*; *van Schaik et al., 2021*), we argue that the relation between ecology, neurobiology, and cognition can be better captured with more specific brain measures, and more specific cognitive operations, than by using the cruder measure of brain size. We believe that comparisons between brain regions have the potential to identify which patterns of brain region evolution can provide insights into the evolution of cognition. Our new set of results are in agreement with a previous study we conducted on another brain region, the VMPFC, critically involved in value-based decision-making (*Louail et al., 2019*). This study was conducted on 29 brain scans from only five species, such that we could not use a PGLS and instead included phylogenetic distance as a co-regressor, along social and ecological variables. As we did here, we also identified the combination of socio-ecological variables that best predicted neuro-anatomical variability across species. Interestingly, the pattern reported for the whole brain was similar to that of the current study, and the weight of the ecological variable (daily traveled distance) was one order of magnitude greater than the influence of the social variable (group size). As it was the case here, VMPFC and whole brain volumes were strongly correlated (r=0.99), but the size of the VMPFC was predicted by a distinct set of ecological variables (dietary quality and weaning age), with little modulation by group size. Thus, our results from these

two combined studies suggest that specific neuro-cognitive entities, established in laboratory studies and defined by a conjunction of specific brain regions and cognitive operations, can be related to specific socio-ecological challenges that animals face in natural conditions.

In conclusion, our results confirm that the size of specific brain regions can be related to socio-ecological variables through the cognitive operations relying on these regions in laboratory conditions. Thus, our approach which aims at articulating cognitive operations reported in laboratory settings with real socio-ecological challenges should provide a clear insight into the neuro-cognitive operations at play in the wild, as well as their evolution in primates. Conversely, integrating realistic socio-ecological challenges can provide a strong insight into the evolution of specific brain functions in primates. Of course, we would need to provide a clearer model regarding how facing socio-ecological challenges can rely upon specific and dynamic cognitive operations, but we believe that this is a critical first step that helps building a common theoretical framework to cross boundaries across behavioral ecology and cognitive neurosciences.

# Materials and methods
## Sample

Thirty-one brain magnetic resonance imaging (MRI) 3D reconstructions from 16 primate species (*Ateles fusciceps*, n=1; *Cebus capucinus*, n=1; *Cercopithecus mitis*, n=1; *Gorilla gorilla*, n=5; *Gorilla beringei*, n=1; *H. sapiens*, n=4; *Hylobates lar*, n=1; *Lagothrix lagotricha*, n=1; *Lophocebus albigena*, n=1; *Macaca fascicularis*, n=1; *Macaca fuscata*, n=4; *Macaca mulatta*, n=2; *Pan troglodytes*, n=5; *Pan paniscus*, n=1; *Papio papio*, n=1, *Pongo pygmaeus*, n=1) were used in this study. Japanese macaques (*M. fuscata*) and rhesus macaques (*M. mulatta*) were captive animals scanned at the National Institutes for Quantum and Radiological Science and Technology (Chiba, Japan) and at Brain and Spine Institute (Paris, France), respectively. *P. troglodytes* and *G. gorilla* brains came from the Muséum national d'Histoire naturelle (Paris, France). They had been collected between 1920 and 1970 and subsequently preserved in formalin solution. The *P. paniscus* and *G. beringei* brains came from the Royal Museum for Central Africa (RMCA) (Tervuren, Belgium) and the Royal Belgian Institute of Natural Sciences (Bruxelles, Belgium). All the *Pan* and *Gorilla* specimens have been scanned at University of Leuven (KUL). The brain scans *A. fusciceps, C. mitis,* and *L. lagotricha* were obtained from the Primate Brain Bank, NIN Utrecht University. Finally, the remaining species (*C. capucinus, H. lar, L. albigena, P. pygmaeus, M. fascicularis, P. papio,* and one specimen of *H. sapiens*) came from the brain catalogue website (https://braincatalogue.org), which gathers scans of specimens from the collections of the Muséum national d'Histoire naturelle (Paris). The three other *H. sapiens* brain scans were obtained from the Allen Institute (online brain atlas). All specimens were sexually mature at the time of scanning. The sexes of individuals were mostly unknown. Moreover, some specimens in the sample came from captivity. Thus, we neglected the effects of captivity and sex on brain/endocranium measurements, which were both shown to be very small compared to inter-species differences (*Isler and van Schaik, 2012*).

## Processing of brain MRI and measurements

Brain measurements (visualization, segmentation, and quantification of brain tissues volumes) were processed using Avizo v9.0 software. The whole brain volume was measured in order to facilitate comparisons with the literature. The cerebellum was excluded from all brain measurements, because it was missing on some of the MRI scans (gorilla brains). Whole brain segmentation was performed using the semi-automated tool in Avizo that enables to select a material or structure according to a specific gray-level threshold. It was however necessary to bring some manual corrections, e.g., when the brain had a similar gray-level than an adjacent tissue. Segmentations of the FP and the DLPFC were carried out manually with the brush tool, slice by slice of the MRI scan.

The FP is the most rostral part of the prefrontal cortex. Cytoarchitectonic studies indicate that it strongly overlaps with BA 10 (*Ongür et al., 2003*; *Ramnani and Owen, 2004*; *Semendeferi et al., 2001*; *Tsujimoto et al., 2011*). Besides cytoarchitectonic landmarks, the FP can also be identified based on connectivity patterns: it receives projections from the temporal superior cortex (*Petrides, 2005*) and has connections with the superior temporal sulcus (*Sallet et al., 2013*). We also used the probabilistic maps of the FP proposed by *John et al., 2007*, and *Bludau et al., 2014*. We delimited

the FP according to different criteria: it should match the functional anatomy for known species (macaques and humans, essentially) and be reliable enough to be applied to other species using macroscopic neuroanatomical landmarks. We integrated these criteria and the data from the literature on brain atlases of rhesus macaques and humans (*Borden et al., 2015*; *Saleem and Logothetis, 2007*) to define visible limits of the FP, as shown in *Figure 1*. The anterior limit of the FP was defined as the anterior limit of the brain. The cingulate sulcus represented the posterior limit. The ventral limit was set as the dorsal limit of the gyrus rectus. Finally, the dorsal limit was defined as the fundus of the superior frontal sulcus in humans and apes, or the principal sulcus in monkeys. We chose this very conservative dorsal limit in monkeys to avoid any overlap with what could be considered as part of the DLPFC (typically, BA 9). However, we are aware that several studies in macaques suggest that the FP in monkeys might extend more dorsally (*Sallet et al., 2013*; *Tsujimoto et al., 2011*). Thus, we also conducted a series of measures with a less conservative dorsal limit for the FP (see *Supplementary file 4*). We used this procedure for 10 species (i.e. the monkeys species), because the delimitation of the FP could be considered from a different perspective. This procedure was not necessary for great apes and humans, because in those species the delimitation of FP was easier. In short, we used a conservative measure of the FP in monkeys, which did not include the dorsal part of the anterior prefrontal cortex. Indeed, even anatomo-functional studies suggest that the the FP could extend more dorsally in macaques, but that dorsal region also includes BA 9, which is usually associated with the DLPFC (*Sallet et al., 2013*; *Petrides, 2005*. *Petrides et al., 2012*). Since that dorsal area could also be associated with the FP, we conducted a new analysis where the FP volume did include that dorsal area, and the details of this analysis are provided in *Supplementary file 4*.

Similarly, the DLPFC was measured by combining functional and anatomical data from the literature to identify reliable macroscopic landmarks (*Levy and Goldman-Rakic, 2000*; *Passingham et al., 2012*; *Passingham and Sakai, 2004*; *Petrides et al., 2012*; *Sallet et al., 2013*). Given the major difference in sulcal organization between monkeys and great apes, we used different landmarks, shown on *Figure 2a and b*. DLPFC comprises portions of middle frontal gyrus and superior frontal gyrus in great apes and lies in and around the principal sulcus in macaques. Then, the ventral limit was set as the fundus of the principal sulcus for monkeys, and the frontal inferior sulcus for apes and humans. The anterior limit of the DLPFC was defined as the posterior limit of the FP, which was the cingulate sulcus. For apes and humans, the posterior limit was defined as the precentral sulcus, whereas in monkeys it was defined as the end of the arcuate sulcus. Finally, the medial limit was designated as the interhemispheric sulcus.

## Socio-ecological and phylogenetic data

Eleven socio-ecological variables were selected for the analyses, gathered in different categories: body condition (body mass), diet (dietary quality index and tool use), movements and ranging behavior (daily traveled distance), social parameters (group size, population density, social system), and variables related to reproduction and life-history traits (mating system, mate guarding, seasonal breeding, and weaning age). Each variable was assessed based on the literature on wild populations, whenever possible, which was the case in a vast majority of cases (and otherwise specified, see below). We verified that these variables showed minimal correlation (see *Supplementary file 7*).

The dietary quality index (DQI) was used to characterize the richness of the dietary spectrum and was calculated from the formula $DQI = 1s + 2r + 3.5a$, where s is the percentage of plant structural parts in the diet, r the percentage of plant reproductive parts, and a the percentage of animal preys (*Sailer et al., 1985*). Thereby, a low index (around 100) characterizes folivorous diets, while a high index (around 200) characterizes more diversified diets (including animals and fruits). Tool use represented the occurrence and complexity of using objects in feeding contexts. We took the definition of *St Amant and Horton, 2008*, which states that 'tool use is the exertion of control over a freely manipulable external object (the tool) with the goal of (1) altering the physical properties of another object, substance, surface, or medium (the target, which may be the tool user or another organism) via a dynamic mechanical interaction, or (2) mediating the flow of information between the tool user and the environment or other organisms in the environment'. It includes notions of control over an object and goal. The presence or absence of tool use in feeding contexts for species in the wild was assessed from literature. Daily traveled distance was expressed in kilometers. Regarding social parameters, group size was defined as the mean social group size of a species. We also used population density

(number of individuals per km²) in order to bring spatial precisions over the group size variable. Group size and population density data were collected and compiled from several primary and secondary sources (see *Supplementary file 8*).

For group size and population density of *H. sapiens*, we took an average between industrialized societies and hunter-gatherers societies. Social system was defined using the four-way categorization scheme typically used in primate studies and included solitary, pair-living, polygyny, and polygynandry (see also *DeCasien et al., 2017*). Mating system categories included spatial polygyny (among solitary species, agonistically powerful males defend mating access to several females), monogamy (one male is socially bonded to one breeding female), polyandry (one female is simultaneously bonded to multiple males), harem polygyny (one male is simultaneously bonded to multiple breeding females), and polygynandry (multiple males and multiple females breed within the same group, but no lasting bonds are formed) (*Clutton-Brock, 1989*). Mate guarding is defined as a female monopolization over an extended period of time to secure paternity (*Manson, 1997*) and is characterized by male attempts to associate and copulate with a female during the presumptive fertile period (*Dixson and Press, 2012*; *Manson, 1997*). This categorical variable was divided into two categories: species using mate guarding and those that do not usually show this behavior. Finally, in order to account for differences in lifespan between study species, the weaning age was calculated as a percentage of maximum lifespan. For some missing values from wild studies, data were taken from studies in captivity and compared to data related to close species (in the wild and in captivity). This method was applied for the weaning age and the dietary quality index of *P. papio*, with other species of baboons. Moreover, the absence of tool use for *L. albigena* was inferred from the absence of published papers or other forms of communication on this subject.

The phylogenetic tree was obtained from the 10ktrees website (https://10ktrees.nunn-lab.org/Primates/downloadTrees.php, version 3). This version (*Arnold et al., 2010*) provides a Bayesian inference of the primate phylogeny based on collected data for 11 mitochondrial and 6 autosomal genes from GenBank across 301 primate species.

## Statistical analysis

We used a PGLS approach to evaluate the joint influence of socio-ecological variables on the neuroanatomical variability across species (*Grafen, 1989*). This approach allowed us to take into account the phylogenetic relation across species when evaluating the influence of socio-ecological variables on their neuroanatomy.

To identify the combination of socio-ecological variables that best predicted the size of a given brain region, given the phylogenetic relations across species, we fitted neuro-anatomical data with several PGLS models, each reflecting a specific combination of socio-ecological variables. Brain measurements and body mass were log10-transformed before analyses. For the same model, several correlation structures were used: standard Brownian, Pagel's (*Pagel, 1999*), and OU-based (*Martins and Hansen, 1997*) and were compared based on the smallest AIC values (*Akaike, 1974*). For a given correlation structure, the effects of the social and ecological variables were assessed with likelihood-ratio tests. All analyses were performed with ape (*Paradis and Schliep, 2019*). We also calculated standardized coefficients, to facilitate comparison. Standardized coefficients were calculated by the product of the coefficient estimated by PGLS with the ratio of the standard deviation of the predictor on the standard deviation of the response (*Supplementary file 2*).

Because of the relatively small number of species in our sample, we assessed the reliability of the inferred models with a 'leave-one-out' procedure: we removed one species from the data and the phylogenetic tree and re-fitted the model selected previously. This was repeated for each species. We conducted this analysis on the best model based on the model comparison with all species included. This model was the same for all three brain measures and it included three variables: body mass, daily traveled distance, and population density (Pop_d). Thus, the model fit was conducted 16 times and each time we evaluated the statistical significance (p-value) for coefficients (beta weights) of each of the three regressors. The specific results of this analysis are provided in *Supplementary files 3, 5, and 6*, for each of the brain measures.

## Acknowledgements

The authors would like to thank Jerome Sallet, Celine Amiez, and Emmanuel Procyk for critical insight into the comparative anatomy of the prefrontal cortex. We also thank Marc Herbin, MNHN, for allowing us to access some of the ape's brains used for this study. This is publication ISEM 2023-152.

## Additional information

### Funding

| Funder | Grant reference number | Author |
|---|---|---|
| CNRS | recurrent founding | Sebastien Bouret<br>Sandrine Prat<br>Cecile Garcia |
| ANR | ANR-17-CE27-0005 (HOMTECH) | Sandrine Prat |

The funders had no role in study design, data collection and interpretation, or the decision to submit the work for publication.

### Author contributions

Sebastien Bouret, Conceptualization, Resources, Data curation, Software, Formal analysis, Supervision, Funding acquisition, Validation, Investigation, Visualization, Methodology, Writing - original draft, Project administration, Writing - review and editing; Emmanuel Paradis, Conceptualization, Resources, Data curation, Software, Formal analysis, Validation, Visualization, Methodology, Writing - review and editing; Sandrine Prat, Conceptualization, Supervision, Funding acquisition, Investigation, Writing - review and editing; Laurie Castro, Formal analysis, Investigation, Methodology; Pauline Perez, Data curation, Formal analysis, Investigation, Writing - review and editing; Emmanuel Gilissen, Resources, Data curation, Writing - review and editing; Cecile Garcia, Conceptualization, Resources, Data curation, Formal analysis, Supervision, Validation, Investigation, Visualization, Methodology, Project administration, Writing - review and editing

### Author ORCIDs

Sebastien Bouret ⓘ https://orcid.org/0000-0003-2279-6161

Reviewer #1 (Public Review): https://doi.org/10.7554/eLife.87780.4.sa1
Author response https://doi.org/10.7554/eLife.87780.4.sa2

## Additional files

### Supplementary files

• Supplementary file 1. Table of AIC values for each predictor of the best model, for each of the brain regions ('response' of the model). AIC values have been computed for a regression model without (OLS) and with (phylogenetic generalized least squares [PGLS]) phylogenetic correlation. The smallest value for each brain region (response) is in bold.

• Supplementary file 2. Standardized coefficients. This table indicates the values of the normalized coefficients for each of the brain region and each of the variables of the regression models.

• Supplementary file 3. Details of the leave-one-out analysis for the whole brain. Each of the three panels corresponds to one of the variables of the model, and shows the value (colored dots) of the estimated coefficient for that variable after the data of the corresponding species has been removed from the analysis. The color code is indicated on the figure, below Pop_d panel. As a reminder, we added the values of these estimated coefficients (together with t and p statistics) when all species are included. The values of these 'original' coefficients (i.e. when all species are included) are indicated by a vertical dotted line on the panel of each of these variables. Body: body weight; DTD: daily traveled distance; Pop_d: population density.

• Supplementary file 4. Alternative analysis with less conservative evaluation of the frontal pole (FP) volume. For 10 species (i.e. the monkeys species), the volume of FP was re-assessed as described in

the text, because the delimitation of the FP could be considered from a different perspective. This procedure was not necessary for great apes and humans, because in those species the delimitation of FP was easier. In short, we used a conservative measure of the FP in monkeys, which did not include the dorsal part of the anterior prefrontal cortex. Indeed, even anatomo-functional studies suggest that the the FP could extend more dorsally in macaques, that dorsal region also includes Brodmann area 9 (BA 9), which is usually associated with the DLPFC (*Sallet et al., 2013*; *Petrides, 2005*. *Petrides et al., 2012*). But since that dorsal area could also be associated with the FP, we conducted a new analysis where the FP volume did include that dorsal area. Fig SF 4.1. Relation between conservative and inclusive measures of the FP volume Table SF 4.2: Model comparison data for the inclusive measure of the FP. Each line corresponds to a model tested to account for the variability of FP volumes across species. Each model is defined as a set of predictors. We provide the AIC of each model using both OLS and PGLS regression methods. For each method, the AIC value of the best model (smallest AIC value) is indicated in bold. Table SI 4.3. Parameter estimate for the best model according to OLS. The table provides the mean beta estimate, its standard error (SE), as well as the corresponding t statistic and p-value estimate for each of the parameters of the best model based on OLS approach (i.e. AIC = 10.22). Table S4.4. Parameter estimate for the best model according to PGLS. The table provides the mean beta estimate, its SE, as well as the corresponding t statistic and p value estimate for each of the parameters of the best model based on PGLS approach (i.e. AIC = 10.25).

• Supplementary file 5. Details of the leave-one-out analysis for the frontal pole (FP). Each of the three panels corresponds to one of the variables of the model, and shows the value (colored dots) of the estimated coefficient for that variable after the data of the corresponding species has been removed from the analysis. The color code is indicated on the figure, below Pop_d panel. As a reminder, we added the values of these estimated coefficients (together with t and p statistics) when all species are included. The values of these 'original' coefficients (i.e. when all species are included) are indicated by a vertical dotted line on the panel of each of these variables. Body: body weight; DTD: daily traveled distance; Pop_d: population density.

• Supplementary file 6. Details of the leave-one-out analysis for the dorso-lateral prefrontal cortex (DLPFC). Each of the three panels corresponds to one of the variables of the model, and shows the value (colored dots) of the estimated coefficient for that variable after the data of the corresponding species has been removed from the analysis. The color code is indicated on the figure, below Pop_d panel. As a reminder, we added the values of these estimated coefficients (together with t and p statistics) when all species are included. The values of these 'original' coefficients (i.e. when all species are included) are indicated by a vertical dotted line on the panel of each of these variables. Body: body weight; DTD: daily traveled distance; Pop_d: population density.

• Supplementary file 7. Relation across socio-ecological variables. There are six continuous and five categorical variables. Table SF7.1. Shows pairwise correlations for all pairs of continuous variables (Pearson correlation coefficients below the diagonal; p-values above the diagonal). If we consider a Bonferroni correction for these tests, the critical p-value becomes 0.05/3=0.003. Legends: Body for body mass; QI for dietary quality index; DTD for daily traveled distance; Gp for group size; Pop_d for population density; weaning for ratio weaning period/lifespan. Fig SF 7.2. Shows Pairwise plots of correlations across these continuous socio-ecological variables. Table SF 7.3. Contingency table of the two correlated categorical variables: social system and mating system. Among the five categorical variables, only one pair was statistically significant (Fisher test, p=0.0009): social system and mating system. These two variables are actually mostly redundant as shown by their contingency table. Fig SF 7.4. Relations between continuous vs categorical variables. When looking at the relationships between a continuous and a categorical variables, social system appeared as related to most continuous variables, especially body mass and weaning. Table SF 7.5. Quantification of the relation between continuous vs categorical variables. We performed an ANOVA on these 30 pairs with the continuous variable as response. Setting the significance threshold at 0.05/30=0.0017, only one fit was statistically significant: social system on body mass. If the response (continuous variable) was log-transformed, only one analysis was significant: social system on group size.

• Supplementary file 8. Construction of the socio-ecological database. Table SF 8.1. Socio-ecological variables. This table indicates the socio-ecological variables of interest for each of the species. BM: body mass (kg); DQI: dietary quality index; TU: tool use (yes/no); DTD: daily traveled distance (km); GS: group size; Pop D: population density (ind/km$^2$); Soc. System: social system; MG: mate guarding (yes/no); Seas. B: seasonal breading (yes/no); Mating syst: Mating system. WL: ratio weaning period/lifespan. Table SF 8.2. References socio-ecological variables. This table shows the references used to calculate the values of each socio-ecological variable of each species. Only

names and dates of publication are provided in the table, but detailed references are provided in the list below.

• MDAR checklist

## Data availability

The sources of brain images are provided in the manuscript (methods section), and the measurements that we made are available on the Dryad website (https://doi.org/10.5061/dryad.qfttdz0s4). Socio-ecological data and their sources are fully available in the manuscript and supporting files.

The following dataset was generated:

| Author(s) | Year | Dataset title | Dataset URL | Database and Identifier |
|---|---|---|---|---|
| Bouret S, Paradis E, Prat S, Castro L, Perez P, Gilissen E, Garcia G | 2024 | Primate PFC comparison: Neuroanatomical measures | https://doi.org/10.5061/dryad.qfttdz0s4 | Dryad Digital Repository, 10.5061/dryad.qfttdz0s4 |

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
