## [Editor Report · eLife assessment]

This **important** study correlates the size of various prefrontal brain regions in primate species with socioecological variables like foraging distance and population density. The evidence presented is **solid** but the approach and conclusions are limited to primates with well-defined gyri.

---

## [Referee Report · Reviewer #1 (Public Review)]

The present study provides a phylogenetic analysis of the size prefrontal areas in primates, aiming to investigate whether relative size of the rostral prefrontal cortex (frontal pole) and dorsolateral prefrontal cortex volume vary according to known ecological or social variables.

I am very much in favor of the general approach taken in this study. Neuroimaging now allows us to obtain more detailed anatomical data in a much larger range of species than ever before and this study shows the questions that can be asked using these types of data. In general, the study is conducted with care, focusing on anatomical precision in definition of the cortical areas and using appropriate statistical techniques, such as PGLS.

I have read the revised version of the manuscript with interest. I commend the authors for including the requested additional analyses. I believe these highlight some of the major debates in the field, such as the relationship between absolute and relative brain size of areas. Providing a full description of the data will help this field be more open about these issues. All too often, debates between different groups focus on narrow anatomical or statistical arguments, and having all the data here is important.

I do not agree with some of the statements of the other reviewers regarding development. Clearly, evolution works for a large part by tinkering (forgive the sense of agency) with development, but that does not mean that looking at the end result cannot provide insights. Ultimately, we will look at both phylogeny and ontogeny within the same framework, but the field is not quite there yet.

As I said before, I do believe this is a positive study. I am happy that we as a field are using imaging data to answer more wider phylogenetic questions. Combining detailed anatomy, big data, and phylogenetic statistical frameworks is an important approach.

---

## [Author Response]

The following is the authors’ response to the previous reviews.

**Public Reviews:**

**Reviewer #1 (Public Review):**
The present study provides a phylogenetic analysis of the size prefrontal areas in primates, aiming to investigate whether relative size of the rostral prefrontal cortex (frontal pole) and dorsolateral prefrontal cortex volume vary according to known ecological or social variables.I am very much in favor of the general approach taken in this study. Neuroimaging now allows us to obtain more detailed anatomical data in a much larger range of species than ever before and this study shows the questions that can be asked using these types of data. In general, the study is conducted with care, focusing on anatomical precision in definition of the cortical areas and using appropriate statistical techniques, such as PGLS.I have read the revised version of the manuscript with interest. I agree with the authors that a focus on ecological vs laboratory variables is a good one, although it might have been useful to reflect that in the title.I am happy to see that the authors included additional analyses using different definitions of FP and DLPFC in the supplementary material. As I said in my earlier review, the precise delineation of the areas will always be an issue of debate in studies like this, so showing the effects of different decisions in vital.

We thank the reviewer for these positive remarks and for these very useful suggestions on the previous version of this article.

I am sorry the authors are so dismissive of the idea of looking the models where brain size and area size are directly compared in the model, rather preferring to run separate models on brain size and area size. This seems to me a sensible suggestion.

We agree with the reviewer 1 and the response of reviewer 3 also made it clear to us of why it was an important issue. We have therefore addressed it more thoroughly this time.

First, we have added a new analysis, with whole brain volume included as covariate in the model accounting for regional volumes, together with the socio-ecological variables of interest. As expected given the very strong correlation across all brain measures (>90%), the effects of all socio-ecological factors disappear for both FP and DLPFC volumes when ‘whole brain’ is included as covariate. This is coherent with our previous analysis showing that the same combination of socio-ecological variables could account for the volume of FP, DLPFC and the whole brain. Nevertheless, the interpretation of these results remains difficult, because of the hidden assumptions underlying the analysis (see below).

Second, we have clarified the theoretical reasons that made us choose absolute vs relative measures of brain volumes. In short, we understand the notion of specificity associated with relative measures, but (1) the interpretation of relative measures is confusing and (2) we have alternative ways to evaluate the specificity of the effects (which are complementary to the idea of adding whole brain volume as covariate).

Our goal here was to evaluate the influence of socio-ecological factors on specific brain regions, based on their known cognitive functions in laboratory conditions (working memory for the DLPFC and metacognition for the frontal pole). Thus, the null hypothesis is that socio-ecological challenges supposed to mobilize working memory and metacognition *do not* affect the size of the brain regions associated with these functions (respectively DLPFC and FP). This is what our analysis is testing, and from that perspective, it seems to us that direct measures are better, because within regions (across species), volumes provide a good index of neural counts (since densities are conserved), which are indicative fo the amount of computational resources available for the region. It is not the case when using relative measures, or when using the whole brain as covariate, since densities are heterogenous across brain regions (e.g. Herculano-Houzel, 2011; 2017, but see below for further details on this).

Quantitatively, the theoretical level of specificity of the relation between brain regions and socio-ecological factors is difficult to evaluate, given that our predictions are based on the cognitive functions associated with DLPFC and FP, namely working memory and metacognition, and that each of these cognitive functions also involved other brain regions. We would actually predict that other brain regions associated with the same cognitive functions as DLPFC or FP also show a positive influence of the same socioecological variables. Given that the functional mapping of cognitive functions in the brain remains debated, it is extremely difficult to evaluate quantitatively how specific the influence of the socio-ecological factors should be on DLPFC and FP compared to the rest of the brain, in the frame of our hypothesis.

Critically, given that FP and DLPFC show a differential sensitivity to population density, a proxy for social complexity, and that this difference is in line with laboratory studies showing a stronger implication of the FP in social cognition, we believe that there is indeed some specificity in the relation between specific regions of the PFC and socioecological variables. Thus, our results as a whole seem to indicate that the relation between prefrontal cortex regions and socio-ecological variables shows a small but significant level of specificity. We hope that the addition of the new analysis and the corresponding modifications of the introduction and discussion section will clarify this point.

Similarly, the debate about whether area volume and number of neurons can be equated across the regions is an important one, of which they are a bit dismissive.

We are sorry that the reviewer found us a bit dismissive on this issue, and there may have been a misunderstanding.

Based on the literature, it is clearly established that *for a given brain region*, area volume provides a good proxy for the number of neurons, and it is legitimate to generalize this relation across species *if neuronal densities* are conserved for the region of interest (see for example Herculano-Houzel 2011, 2017 for review). It seems to be the case across primates because cytoarchitectonic maps are conserved for FP and DLPFC, at least in humans and laboratory primates (Petrides et al, 2012; Sallet et al, 2013; Gabi et al, 2016; Amiez et al, 2019). But we make no claim about the difference in number of neurons between FP and DLPFC, and we never compared regional volumes across regions (we only compared the influence of socio-ecological factors on each regional volume), so their difference in cellular density is not relevant here. As long as the neuronal density is conserved *across* species but *within* a region (DLPFC or FP), the difference in volume for that region, across species, does provide a reliable proxy for the influence of the socioecological regressor of interest (across species) on the number of neurons in that region.

Our claims are based on the strength of the relation between (1) cross-species variability in a set of socio-ecological variables and (2) cross-species variability in neural counts in each region of interest (FP or DLPFC). Since the effects of interest relate to inter-specific differences, within a region, our only assumption is that the neural densities are conserved across distinct species for a given brain region. Again (see previous paragraph), there is reasonable evidence for that in the literature. Given that assumption, regional volumes (across species, for a given brain region) provide a good proxy for the number of neurons. Thus, the influence of a given socio-ecological variable on the interspecific differences in the volume of a single brain region provides a reliable estimate of the influence of that socio-ecological variable on the number of neurons in that region (across species), and potentially of the importance of the cognitive function associated with that region in laboratory conditions. None of our conclusions are based on direct comparison of volumes across regions, and we only compared the influence of socioecological factors (beta weights, after normalization of the variables).

Note that this is yet another reason for not using relative measures and not including whole brain as covariate in the regression model: Given that whole brain and any specific region have a clear difference in density, and that this difference is probably not conserved across species, relative measures (or covariate analysis) cannot be used as proxies for neuronal counts (e.g. Herculano-Houzel, 2011). In other words, using the whole brain to rescale individual brain regions relies upon the assumption that the ratios of volumes (specific region/whole brain) are equivalent to the ratios of neural counts, which is not valid given the differences in densities.

Nevertheless, I think this is an important study. I am happy that we are using imaging data to answer more wider phylogenetic questions. Combining detailed anatomy, big data, and phylogenetic statistical frameworks is a important approach.

We really thank the reviewer for these positive remarks, and we hope that this study will indeed stimulate others using a similar approach.

**Reviewer #2 (Public Review):**
In the manuscript entitled "Linking the evolution of two prefrontal brain regions to social and foraging challenges in primates" the authors measure the volume of the frontal pole (FP, related to metacognition) and the dorsolateral prefrontal cortex (DLPFC, related to working memory) in 16 primate species to evaluate the influence of socio-ecological factors on the size of these cortical regions. The authors select 11 socio-ecological variables and use a phylogenetic generalized least squares (PGLS) approach to evaluate the joint influence of these socio-ecological variables on the neuro-anatomical variability of FP and DLPFC across the 16 selected primate species; in this way, the authors take into account the phylogenetic relations across primate species in their attempt to discover the the influence of socio-ecological variables on FP and DLPF evolution.The authors run their studies on brains collected from 1920 to 1970 and preserved in formalin solution. Also, they obtained data from the Mussée National d´Histoire Naturelle in Paris and from the Allen Brain Institute in California. The main findings consist in showing that the volume of the FP, the DLPFC, and the Rest of the Brain (ROB) across the 16 selected primate species is related to three socio-ecological variables: body mass, daily traveled distance, and population density. The authors conclude that metacognition and working memory are critical for foraging in primates and that FP volume is more sensitive to social constraints than DLPFC volume.The topic addressed in the present manuscript is relevant for understanding human brain evolution from the point of view of primate research, which, unfortunately, is a shrinking field in neuroscience. But the experimental design has two major weak points: the absence of lissencephalic primates among the selected species and the delimitation of FP and DLPFC. Also, a general theoretical and experimental frame linking evolution (phylogeny) and development (ontogeny) is lacking.

We are sorry that the reviewer still believes that these two points are major weaknesses.

- We have added a point on lissencephalic species in the discussion. In short, we acknowledge that our work may not be applied to lissencephalic species because they cannot be studied with our method, but on the other hand, based on laboratory data there is no evidence showing that the functional organization of the DLPFC and FP in lissencephalic primates is radically different from that of other primates (Dias et al, 1996; Roberts et al, 2007; Dureux et al, 2023; Wong et al, 2023). Therefore, there is no a priori reason to believe that not including lissencephalic primates prevents us from drawing conclusions that are valid for primates in general. Moreover, as explained in the discussion, including lissencephalic primates would require using invasive functional studies, only possible in laboratory conditions, which would not be compatible with the number of species (>15) necessary for phylogenetic studies (in particular PGLS approaches). Finally, as pointed out by the reviewer, our study is also relevant for understanding human brain evolution, and as such, including lissencephalic species should not be critical to this understanding.

- In response to the remarks of reviewer 1 on the first version of the manuscript, we had included a new analysis in the previous version of the manuscript, to evaluate the validity of our functional maps given another set of boundaries between FP and DLPFC. But one should keep in mind that our objective here is not to provide a definitive definition of what the regions usually referred to as DLPFC and FP should be from an *anatomical* point of view. Rather, as our study aims at taking into account the phylogenetic relations across primate species, we chose landmarks that enable a comparison of the volume of cortex involved in metacognition (FP) and working memory (DLPFC) across species. We have also updated the discussion accordingly.

We agree that this is a difficult point and we have always acknowledged that this was a clear limitation in our study. In the light of the functional imaging literature in humans and non-human primates, as well as the neurophysiological data in macaques, defining the functional boundary between FP and DLPFC remains a challenging issue even in very well controlled laboratory conditions. As mentioned by reviewer 1, “the precise delineation of the areas will always be an issue of debate in studies like this, so showing the effects of different decisions in vital”. Again, an additional analyses using different boundaries for FP and DLPFC was included in the supplementary material to address that issue. Now, we are not aware of solid evidence showing that the boundaries that we chose for DLPFC vs FP were wrong, and we believe that the comparison between 2 sets of measures as well as the discussion on this topic should be sufficient for the reader to assess both the strength and the limits of our conclusion. That being said, if the reviewer has any reference in mind showing better ways to delineate the functional boundary between FP and DLPFC in primates, we would be happy to include it in our manuscript.

- The question of development, which is an important question per se, is neither part of the hypothesis nor central for the field of comparative cognition in primates. Indeed, major studies in the field do not mention development (e.g. Byrne, 2000; Kaas, 2012; Barton, 2012). De Casien et al (2022) even showed that developmental constraints are largely irrelevant (see Claim 4 of their article): [« The functional constraints hypothesis […] predicts more complex, ‘mosaic’ patterns of change at the network level, since brain structure should evolve adaptively and in response to changing environments. It also suggests that ‘concerted’ patterns of brain evolution do not represent conclusive evidence for developmental constraints, since allometric relationships between developmentally linked or unlinked brain areas may result from selection to maintain functional connectivity. This is supported by recent computational modeling work [81], which also suggests that the value of mosaic or concerted patterns may fluctuate through time in a variable environment and that developmental coupling may not be a strong evolutionary constraint. Hence, the concept of concerted evolution can be decoupled from that of developmental constraints »].

Finally, when studies on brain evolution and cognition mention development, it is generally to discuss energetic constraints rather than developmental mechanisms per se (Heldstab et al 2022 ; Smaers et al, 2021; Preuss & Wise, 2021; Dunbar & Schutz, 2017; MacLean et al, 2012. Mars et al, 2018; 2021). Therefore, development does not seem to be a critical issue, neither for our article nor for the field.

**Reviewer #3 (Public Review):**
This is an interesting manuscript that addresses a longstanding debate in evolutionary biology - whether social or ecological factors are primarily responsible for the evolution of the large human brain. To address this, the authors examine the relationship between the size of two prefrontal regions involved in metacognition and working memory (DLPFC and FP) and socioecological variables across 16 primate species. I recommend major revisions to this manuscript due to: (1) a lack of clarity surrounding model construction; and (2) an inappropriate treatment of the relative importance of different predictors (due to a lack of scaling/normalization of predictor variables prior to analysis).

We thank the reviewer for his/her remarks, and for the clarification of his /her criticism regarding the use of relative measures. We are sorry to have missed the importance of this point in the first place. We also thank the reviewer for the cited references, which were very interesting and which we have included in the discussion. As the reviewer 1 also shared these concerns, we wrote a detailed response to explain how we addressed the issue above.

First, we did run a supplementary analysis where whole brain volume was added as covariate, together with socio-ecological variables, to account for the volume of FP or DLPFC. As expected given the very high correlation across all 3 brain measures, none of the socio-ecological variables remained significant. We have added a long paragraph in the discussion to tackle that issue. In short, we agree with the reviewer that the specificity of the effects (on a given brain region vs the rest of the brain) is a critical issue, and we acknowledge that since this is a standard in the field, it was necessary to address the issue and run this extra-analysis. But we also believe that specificity could be assessed by other means: given the differential influence of ‘population density’ on FP and DLPFC, in line with laboratory data, we believe that some of the effects that we describe do show specificity. Also, we prefer absolute measures to relative measures because they provide a better estimate of the corresponding cognitive operation, because standard allometric rules (i.e., body size or whole brain scaling) may not apply to the scaling and evolution of FP and DLPFC in primates.. Indeed, given that we use these measures as proxies of functions (metacognition for FP and working memory for DLPFC), it is clear that other parts of the brain should show the same effect since these functions are supported by entire networks that include not only our regions of interest but also other cortical areas in the parietal lobe. Thus, the extent to which the relation with socio-ecological variables should be stronger in regions of interest vs the whole brain depends upon the extent to which other regions are involved in the same cognitive function as our regions of interest, and this is clearly beyond the scope of this study. More importantly, volumetric measures are taken as proxies for the number of neurons, but this is only valid when comparing data from the same brain region (across species), but not across brain regions, since neural densities are not conserved. Thus, using relative measures (scaling with the whole brain volume) would only work if densities were conserved across brain regions, but it is not the case. From that perspective, the interpretation of absolute measures seems more straightforward, and we hope that the specificity of the effects could be evaluated using the comparison between the 3 measures (FP, DLPFC and whole brain) as well as the analysis suggested by the reviewer. We hope that the additional analysis and the updated discussion will be sufficient to cover that question, and that the reader will have all the information necessary to evaluate the level of specificity and the extent to which our findings can be interpreted.

**Recommendations for the authors:**

**Reviewer #2 (Recommendations For The Authors):**
In my previous review of the present manuscript, I pointed out the fact that defining parts, modules, or regions of the primate cerebral cortex based on macroscopic landmarks across primate species is problematic because it prevents comparisons between gyrencephalic and lissencephalic primate species. The authors have rephrased several paragraphs in their manuscript to acknowledge that their findings do apply to gyrencephalic primates.I also said that "Contemporary developmental biology has showed that the selection of morphological brain features happens within severe developmental constrains. Thus, the authors need a hypothesis linking the evolutionary expansion of FP and DLPFC during development. Otherwise, the claims form the mosaic brain and modularity lack fundamental support". I insisted that the author should clarify their concept of homology of cerebral cortex parts, modules, or regions cross species (in the present manuscript, the frontal pole and the dorsolateral prefrontal cortex). Those are not trivial questions because any phylogenetic explanation of brain region expansion in contemporary phylogenetic and evolutionary biology must be rooted in evolutionary developmental biology. In this regard, the authors could have discussed their findings in the frame of contemporary studies of cerebral cortex evolution and development, but, instead, they have rejected my criticism just saying that they are "not relevant here" or "clearly beyond the scope of this paper".

The question of development, which is an important question per se, is neither part of the hypothesis nor central for the field of comparative cognition in primates. Indeed, the major studies in the field do not mention development and some even showed that developmental constraints were not relevant (see De Casien et al., 2022 and details in our response to the public review). When studies on brain evolution and cognition mention development, it is generally to discuss energetic constraints rather than developmental mechanisms per se (Heldstab et al 2022 ; Smaers et al, 2021; Preuss & Wise, 2021; Dunbar & Schutz, 2017; MacLean et al, 2012. Mars et al, 2018; 2021).

If the other reviewers agree, the authors are free to publish in eLife their correlations in a vacuum of evolutionary developmental biology interpretation. I just disagree. Explanations of neural circuit evolution in primates and other mammalian species should tend to standards like the review in this link: https://royalsocietypublishing.org/doi/full/10.1098/rstb.2020.0522

In this article, Paul Cizek (a brilliant neurophysiologist) speculates on potential evolutionary mechanisms for some primate brain functions, but there is surprisingly very little reference to the existing literature on primate evolution and cognition. There is virtually no mention of studies that involve a large enough number of species to address evolutionary processes and/or a comparison with fossils and/or an evaluation of specific socio-ecological evolutionary constraints. Most of the cited literature refers to laboratory studies on brain anatomy of a handful of species, and their relevance for evolution remains to be evaluated. These ideas are very interesting and they could definitely provide an original perspective on evolution, but they are mostly based on speculations from laboratory studies, rather than from extensive comparative studies. This paper is interesting for understanding developmental mechanisms and their constraints on neurophysiological processes in laboratory conditions, but we do not think that it would fit it in the framework of our paper as it goes far beyond our main topic.

**Reviewer #3 (Recommendations For The Authors):**
Yes, I am suggesting that the authors also include analyses with brain size (rather than body size) as a covariate to evaluate the effects of other variables in the model over and above the effect on brain size. In a very simplified theoretical scenario: two species have the same body sizes, but species A has a larger brain and therefore a larger FP. In this case, species A has a larger FP because of brain allometric patterns, and models including body size as a covariate would link FP size and socioecological variables characteristic of species A (and others like it). However, perhaps the FP of species A is actually smaller than expected for its brain size, while the FP of species B is larger than expected for its brain size.

As explained in our response to the public review, we did run this analysis and we agree with the reviewer’s point from a practical point of view: it is important to know the extent to which the relation with a set of socio-ecological variables is specific of the region of interest, vs less specific and present for other brain regions. Again, we are sorry to not have understood that earlier, and we acknowledge that since it is a standard in the field, it needs to be addressed thoroughly.

We understand that the scaling intuition, and the need to get a reference point for volumetric measures, but here the volume of each brain region is taken as a proxy for the number of neurons and therefore for the region’s computational capacities. Since, for a given brain region (FP or DLPFC) the neural densities seem to be well conserved across species, comparing regional volumes across species provides a good proxy for the contrast (across species) in neural counts for that region. All we predicted was that for a given brain region, associated with a given cognitive operation, the volume (number of neurons) would be greater in species for which socio-ecological constraints potentially involving that specific cognitive operation were greater. We do not understand how or why the rest of the brain would change this interpretation (of course, as discussed just above, beyond the question of specificity). And using whole brain volume as a scaling measure is problematic because the whole brain density is very different from the density of these regions of the prefrontal cortex (see above for further details). Again, we acknowledge that allometric patterns exist, and we understand how they can be interpreted, but we do not understand how it could prove or disprove our hypothesis (brain regions involved in specific cognitive operations are influenced by a specific set of socio-ecological variables). When using volumes as a proxy for computational capacities, the theoretical implications of scaling procedures might be problematic. For example, it implies that the computational capacities of a given brain region are scaled by the rest of the brain. All other things being equal, the computational capacities of a given brain region, taken as the number of neurons, should decrease when the size of the rest of the brain increases. But to our knowledge there is no evidence for that in the literature. Clearly these are very challenging issues, and our position was to take absolute measures because they do not rely upon hidden assumptions regarding allometric relations and their consequence on cognition.

But since we definitely understand that scaling is a reference in the field, we have not only completed the corresponding analysis (including the whole brain as a covariate, together with socio-ecological variables) but also expended the discussion to address this issue in detail. We hope that between this new analysis and the comparison of effects between non-scaled measures of FP, DLPFC and the whole brain, the reader will be able to judge the specificity of the effect.

Models including brain (instead of body) size would instead link FP size and socioecological variables characteristic of species B (and others like it). This approach is supported by a large body of literature linking comparative variation in the relative size of specific brain regions (i.e., relative to brain size) to behavioral variation across species - e.g., relative size of visual/olfactory brain areas and diurnality/nocturnality in primates (Barton et al. 1995), relative size of the hippocampus and food caching in birds (Krebs et al. 1989).Barton, R., Purvis, A., & Harvey, P. H. (1995). Evolutionary radiation of visual and olfactory brain systems in primates, bats and insectivores. Philosophical Transactions of the Royal Society of London. Series B: Biological Sciences, 348(1326), 381-392.Krebs, J. R., Sherry, D. F., Healy, S. D., Perry, V. H., & Vaccarino, A. L. (1989). Hippocampal specialization of food-storing birds. Proceedings of the National Academy of Sciences, 86(4), 1388-1392.

We are grateful to the reviewer for mentioning these very interesting articles, and more generally for helping us to understand this issue and clarify the related discussion. Again, we understand the scaling principle but the fact that these methods provide interesting results does not make other approaches (such as ours) wrong or irrelevant. Since we have used both our original approach and the standard version as requested by the reviewer, the reader should be able to get a clear picture of the measures and of their theoretical implications. We sincerely hope that the present version of the paper will be satisfactory, not only because it is clearer, but also because it might stimulate further discussion on this complex question.